# The Role of Dietary Fiber in Improving Pig Welfare

**DOI:** 10.3390/ani13050879

**Published:** 2023-02-28

**Authors:** Sungho Do, Jae-Cheol Jang, Geon-Il Lee, Yoo-Yong Kim

**Affiliations:** 1Department of Agricultural Biotechnology and Research Institute of Agriculture and Life Sciences, Seoul National University, 1, Gwanak-ro, Gwanak-gu, Seoul 08826, Republic of Korea; 2Department of Animal Science, Gyeongsang National University, 33 Dongjin-ro, Gyeonsangnam-do, Jinjusi 52725, Republic of Korea; 3Division of Animal Science, Chonnam National University, 77, Yongbong-ro, Buk-gu, Gwangju 61186, Republic of Korea

**Keywords:** fiber, stereotypies, animal welfare, sow, gestation

## Abstract

**Simple Summary:**

The restriction of feed intake in sows during the gestation period is important for the livestock producer due to the prevention of excess body weight gain and fat deposition, leading to low reproductive performance and detrimental effects at farrowing and during lactation. The frustration caused by feed restriction is the major factor in developing stereotypic behaviors in sows and the occurrence of this behavior may be associated with sows being hungry. Therefore, several strategies have been employed to ameliorate the stereotypic behaviors in sows. Feeding sows a high-fiber diet is the most effective method to increase postprandial satiety, thereby improving the welfare of sows subjected to feed restriction during pregnancy. There are a number of fibrous ingredients available, including wheat middlings, corn germ, sugar beet pulp, corn gluten feed, soy hulls, dried grass, and alfalfa meal. However, the effect of fiber-rich diets on satiety and behavior depends on the dietary fiber sources (physicochemical properties) or fiber inclusion rate in the diet. The objective of this review is to discuss the functional roles of dietary fiber sources with different inclusion levels reducing abnormal behaviors in sows.

**Abstract:**

This review aims to discuss the effects of dietary fiber sources with various levels on stereotypic behaviors in sows. There are a variety of dietary fiber sources that are supplemented to feeds for sows. However, dietary fiber sources have different physio-chemical properties, leading to controversial results in feed motivation, nutrient digestibility, and behaviors in sows fed fiber-rich diets. Findings from previous studies indicated that soluble fiber delays nutrient absorption and decreases physical activity after feeding. In addition to this, it increases volatile fatty acid production, provides energy, and prolongs the feeling of satiety. It also prevents certain stereotypies and thus is paramount to sow welfare.

## 1. Introduction

Animal welfare has been becoming an important area of concern for livestock producers over the last three decades. Public concern over the methods used to raise food-producing animals has increased, and these concerns are leading to voluntary and mandated changes in the methods used to produce livestock [1]. The primary aim of these changes in production methods is to improve the welfare of livestock.

Accurately determining the welfare of animals is difficult [2] and should employ a multidimensional approach [3]. Multiple factors (e.g., animal health, immune system competence, housing, behavior, growth performance, reproduction, environmental conditions, and physiological characteristics) should be taken into consideration for the assessment of animal welfare [3,4,5]. Most people intuitively assume that hunger and welfare are negatively correlated. Hunger is a relatively easy concept to understand, but a difficult sensation to measure in animals. There are many factors (mixing of pigs different origin, nutrient deficient, high stocking density, boredom, temperature variation, stress, discomfort, or pain) that contribute to animal welfare, and the presence of hunger could not necessarily mean that welfare is compromised. Further research and advances in animal welfare assessment are necessary to shed greater light on the relationship between hunger and welfare.

It is possible to measure hunger directly through operant conditioning tests [6] and indirectly through stereotypical behaviors and postures of the sows [7]. Operant conditioning tests are likely to give the most accurate results, but they are not easily available and amenable to production conditions. Therefore, in the current discussion, stereotypical behaviors and the sow posture will be considered as markers of hunger. Stereotypic behaviors are actions that are frequently repeated, have no apparent purpose, and appear to be useless to the animal. According to [8], stereotypical actions can include, but are not limited to, biting on bars, sham or vacuum chewing (chewing motions unrelated to eating), and nosing or licking the floor or feeder in the absence of food (Table 1). In addition to stereotypic behaviors, the proportion of time spent standing and/or active as opposed to lying (Table 1) is used to indicate hunger because sows do not appear satiated [7].

We acknowledge that using simplified ways to assess a complicated physiological process such as eating motivation might be misleading [11]. However, in production, basic approaches are now the only viable tools available. Stereotypic behaviors are more common in sows, and they are closely related to the restricted feeding levels used in commercial production [12], which supports the idea that stereotypies might be used as a sign of hunger and feeding motivation [13]. The level of feeding and associated satiety experienced by sows appears to be inversely related to the duration of sow stereotypies [14]. Interestingly, stereotypic behavior can be expressed in restrictively-fed gilts housed individually or in groups [12], indicating the importance of feeding level to animal welfare.

## 2. Approaches to Mitigate Hunger and Stereotypic Behaviors

### 2.1. Higher Feeding Levels

An obvious approach to ameliorating hunger and associated stereotypes is to offer pregnant sows a much higher quantity of feed. Increasing the amount of nutrient-dense diets used in commercial production effectively satiates sows and reduces stereotypic behavior. Unfortunately, this approach also supports excessive maternal weight gain during pregnancy, which suppresses feed intake during the subsequent lactation [15] and compromises sow longevity [16]. Furthermore, increased feeding of nutrient-dense diets increases production costs and results in sows with a large body size that do not easily fit in existing accommodations [2,17]. It is unlikely that this approach alone will alleviate hunger in gestating sows.

Since it is well known that an increase in feed intake plays an important role in decreasing stereotypical behavior, many researchers have concentrated on providing diets that are relatively low in nutrient density and bulk density in order to decrease stereotypical behavior [14,18,19]. Consequently, low nutrient-dense diets can be fed to sows at relatively higher levels without excessive gains in sow body weight. Typically, low-density diets contain a high proportion of fibrous feed ingredients. Examples of fibrous feed ingredients commonly used to dilute the nutrient density of diets include sugar beet pulp, soybean hulls, wheat bran, oat hulls, alfalfa meal, alfalfa hay, and wheat bran. One must carefully choose the fibrous ingredient for use in sow diets because they differ dramatically in nutrient composition and digestibility.

### 2.2. Feeding a High-Fiber Diet

#### 2.2.1. Dietary Fiber

Dietary fiber has been used to describe the plant-derived component of feed and foods, which was isolated or synthetic non-digestible carbohydrates and resistant to digestion by mammalian enzymes [15,20]. Dietary fiber could be categorized by source, chemical characteristics, resistance to digestion, and beneficial physiological effects. However, the simple way to classify with dietary fiber is to be divided into two primary classes: soluble dietary fiber and insoluble dietary fiber (Table 2) [21,22,23]. Simply stated, they are classified based on their ability to dissolve in water. Soluble dietary fiber typically includes compounds such as mix-linkage glucans, arabinoxylan, gums, and oligosaccharides including fructooligosaccharide, pectins, and β-glucans (Table 3). On the other hand, insoluble dietary fiber contains cellulose, hemicelluloses, lignin, resistant starches, and polyphenols (Table 3). Both soluble and insoluble dietary fiber are found in feed ingredients for pigs (Table 2). However, the solubility of dietary fiber in feedstuff varies depending on botanical origin of the plants, processing methods, and harvest conditions [21,24,25].

Dietary fiber has many different functions and activities as it passes through the gastrointestinal tract. For example, soluble dietary fiber could delay gastric emptying and may slow nutrient uptake, while insoluble dietary fiber stimulates the production of beneficial bacteria in the colon, increases the passage rate, and alters the digestive enzyme activity [15,26,27,28,29]. Based on these functions of dietary fiber (Table 3), they could be thought to impact on the satiation for sows because of their properties of adding bulking and viscosity in the gut.

The use of dietary fiber for sow is a reasonable method to reduce hunger and stereotypies. While fiber is a carbohydrate, it is not easily digestible. This means it could provide feelings of fullness after feeding without spiking blood sugar or providing extra calories [21,26]. In addition, dietary fiber mitigates the symptom of constipation, thereby alleviating stress because pregnant sows are often subject to feeding restriction. Although the addition of dietary fiber plays an important role in alleviating sow hunger and reducing abnormal behavior, it may be affected by the type of fiber and the inclusion level in the diet [20,26,27,28,29].

**Table 2 animals-13-00879-t002:** Fiber composition of feed ingredients used in pig diets ^a^.

Ingredients	Type of Fiber, %		Detergent Method, %		
	Soluble	Insoluble	ADF	NDF	ADL
Barely	5.4	9.7	5.8	18.3	2.3
Corn	0.9	6.0	2.9	9.1	0.3
Corn DDGS	3.0	14.1	12.0	30.5	2.6
Oat, whole	3.6	9.8	13.7	25.3	-
Oat hulls	4.9	65.7	32.1	65.9	5.4
Rye	3.7	8.4	4.6	12.3	0.8
Sorghum	0.6	5.1	4.9	10.6	0.4
Sorghum hulls	10.0	45.0	41.6	59.4	-
Sugar beet pulp	25.2	18.0	23.5	44.9	-
Wheat	2.3	6.8	3.6	10.6	1.0
Wheat bran	2.5	23.8	11.0	32.3	-

^a^ Reprinted/adapted with permission from Ref. [30]. 2012, NRC.; Reprinted/adapted with permission from Ref. [31]. 2015, Jha and Berrocoso; Reprinted/adapted with permission from Ref. [32]. 2016, Jiménez-Moreno et al.

**Table 3 animals-13-00879-t003:** Characteristics of dietary fiber and relationships to gut effects ^a^.

Dietary fiber	Solubility	GIT Effect ^b^
Cellulose	Insoluble	Substrate for microbial fermentation Changes digesta mixing Alteration of digestive enzyme activity Stimulates passage rate
Lignin	Insoluble	
Resistant starches	Insoluble	
Hemicellulose	Insoluble	
Polyphenols	Insoluble	
Mix-linkage glucans	Soluble	Changes in mixing and diffusion Slows gastric emptying and glucose absorption Increases viscosity in the upper GIT
Arabinoxylan	Soluble	
β-glucan	Soluble	
Pectins	Soluble	
Gums	Soluble	
Non-digestible oligosaccharides	Soluble	
Fructooligosaccharide	Soluble	

^a^ Reprinted/adapted with permission from Ref. [33]. 2000, Jimenez-Escrig and Sanchez-Muniz; Reprinted/adapted with permission from Ref. [26]. 2007, Maljaars et al.; Reprinted/adapted with permission from Ref. [27]. 2010, Gunness and Gidley; Reprinted/adapted with permission from Ref. [15]. 1994, Low et al.; Reprinted/adapted with permission from Ref. [28]. 2015, Zhang et al.; Reprinted/adapted with permission from Ref. [29]. 2016, Gunness et al.; Reprinted/adapted with permission from Ref. [21]. 2019, Williams et al. ^b^ Gastrointestinal tract effect.

#### 2.2.2. Sources of Dietary Fiber

When fibrous ingredients are incorporated into the diet of sows, the carbohydrate composition would be changed from a high-starch diet to a less-starch diet (more non-starch polysaccharides). In general, dietary fiber has a lower nutritional value than other ingredients, but the ingestion of high-fiber diets has a beneficial effect on animal welfare. Minimizing stereotypies in sows during the gestation period requires providing sufficient feed to satisfy feed motivation. The bulk offered by a diet rich in fibrous ingredients could reduce the abnormal behaviors in sows without excessive fatness and reduce reproductive performance. Different sources and types of dietary fiber have been used to evaluate how those fiber sources mitigate symptoms of feed motivation and stereotypies in sows. However, the results have been controversial [27,34,35,36,37].

Generally, the inclusion of dietary fiber in the diet may reduce appetite and voluntary feed intake. It is not clear, however, whether a lowered palatability or gut fill is the major reason for reduced feed intake. Krogh et al. [35] reported that feeding high-fiber diets based on sugar beet pulp (12%) or alfalfa meal (17%) to sows during the lactation period had no differences in feed intake, weight gain, and backfat thickness. Similarly, no differences were observed in feed intake and weight gain when sows were fed the diets with 5% of resistant starch and fermented soybean fiber [38]. However, feeding sows with a konjac flour (2%) during gestation increased the subsequent lactation feed intake of sows [34] compared to a control diet. Additionally, increased feed intake was observed in sows fed the diet with wheat straw (12%) and sugar beet pulp (16%) compared to the control diet [39]. Moreover, sows fed sugar beet pulp (20%) increased their feed intake compared to those fed control diets [36]. Weng [40] tested diets containing 20% wheat bran, soya hull, and rice hull for sows. In that study, sows fed the diet with rice hull showed higher weight gain during gestation and consumed more feed during lactation. Similarly, Feyera et al. [37] tested four different fiber sources (mixed fiber, palm kernel expellers, sugar beet pulp, and soy hull) for sows. In this study, sows fed mixed fiber and palm kernel expellers had higher average daily feed intake compared to sows fed sugar beet pulp, and soy hull (Table 4). These results indicated that wheat straw, sugar beet pulp (16% or 20%), konjac flour, rice hull, palm kernel expellers, and mixed fiber showed relatively higher feed intakes compared to other fiber sources. The most likely explanation may be that dietary fiber sources could have considerably different physiological functions between feedstuff, and this may affect the feed intake and weight gain of sows.

Inclusion of fibrous feed ingredients in the diet of gestating sows does not necessarily depress nutrient digestibility. In vivo digestibility of fibrous feed ingredients is usually determined in young growing pigs. However, it is clear that sows have a greater capacity to extract energy from fibrous feedstuffs compared with growing pigs [41]. So, one must carefully extrapolate the digestibility data determined in growing pigs to sows using regression equations or by predicting the energy content of feeds from chemical analysis [41]. Generally, the nutritional value of ingredients with high levels of soluble fiber is greater than ingredients with elevated levels of insoluble fiber because soluble fiber is more completely fermented in the gastrointestinal tract of sows [41].

According to Renteria-Flores [39], who fed pregnant sows diets that contained high levels of soluble, insoluble, or a combination of soluble and insoluble fiber, the diet high in soluble fiber from oat bran was superior in energy digestibility and similar in nitrogen digestibility compared to the low-fiber control diet. However, high levels of insoluble fiber from wheat straw and high levels of both soluble and insoluble fiber from sugar beet pulp depressed the digestibility of energy and nitrogen. Similarly, Feyera et al. [37] supplemented dietary fiber sources such as mixed fiber, palm kernel expellers, sugar beet pulp, and soy hull in the gestational diet for sows. Sows had a similar amount of dietary fiber supplementations during the gestation period. The feed supplemented with soy hull and sugar beet pulp was highest in energy digestibility, while palm kernel expellers and mixed fiber showed lowest in protein and non-starch polysaccharides digestibility. However, sows fed the diet supplemented with 2% of konjac flour had higher neutral detergent fiber and crude protein digestibility compared to the control diet, whereas there were no differences in the digestibility of gross energy, acid detergent fiber, crude fiber, and dry matter ([34], Table 4). The degree of nutrient digestibility is dependent on the source of dietary fiber. In this context, feeding soluble dietary fiber sources to sows increases nutrient digestibility more than sows fed insoluble dietary fiber sources. Ingestion of soluble dietary fiber increases digesta viscosity and thereby slow down the digesta passage rate, allowing more time to nutrient absorption.

**Table 4 animals-13-00879-t004:** Results of studies evaluating the effect of dietary fiber sources on feed intake, nutrient digestibility, and behaviors of sows.

Feeding Periods	Fiber Sources	Ingredient Concentrations	Main Results	References
Gestation	Control vs. oat bran	34%	No effect on N digestibility and feed intake, but increased energy digestibility	Renteria-Flores et al. [39]
	Control vs. wheat straw	12%	Decreased N and energy digestibility, increased feed intake	
	Control vs. sugar beet pulp	16%	Decreased N and energy digestibility, increased feed intake	
Gestation	Control vs. konjac flour	2%	Increased CP and NDF digestibility and lactation feed intake ^a^	Sun et al. [34]
Lactation	Control vs. alfalfa meal	17%	No effect on feed intake, weight gain, and backfat thickness	Krogh et al. [35]
	Control vs. sugar beet pulp	12%		
Gestation	Control vs. sugar beet pulp	20%	Increased feed intake	Shang et al. [36]
	Control vs. wheat bran	30%	No effect on feed intake	
Lactation	Wheat bran vs. soya hull vs. rice hull	20% in all diets	Rice hull showed higher BW gain during gestation and feed intake during lactation than other fiber sources	Weng [40]
Gestation	MF vs. SBP vs. PKE vs. SH ^b^	Supplemented in gestation diet in same level	SH and SBP showed the highest energy digestibility and lowest for PKE	Feyera et al. [37]
			PKE showed the lowest protein digestibility, MF showed the lowest NSP digestibility ^c^	
			Increased ADFI in MF and PKE compared to SBP and SH	
Gestation	Control vs. resistant starch	11%	Lowest time of fighting frequency	Sapkota et al. [42]
	Control vs. sugar beet pulp	27%	Increased percentage of standing behavior	
	Control vs. soybean hulls	19%	Highest percentage of resting behavior	
Gestation and lactation	Control vs. resistant starch	5%	Decreased the time on standing behavior, no difference in feed intake and BW	Huang et al. [38]
	Control vs. fermented soybean fiber	5%	No differences in behaviors (lysing, sitting, licking, chewing, drinking), no difference in feed intake and BW	
Gestation	MIDD-SY vs. DDGS-GM ^d^	30% of wheat middlings and 15% of soybean hulls vs. 30% of distillers dried grains and 30% of corn germ meal	Sows fed MIDD-SY increased percentage of eating behavior and decreased percentage of sitting behavior	Lopez et al. [43]

^a^ CP: crude protein, NDF: neutral detergent fiber. ^b^ Mixed fiber: MF, sugar beet pulp: SBP, palm kernel expellers: PKE, and soy hulls: SH. ^c^ NSP: non-starch polysaccharides. ^d^ MIDD-SY: 30% wheat middlings and 15% soybean hulls, DDGS-GM: 30% distillers dried grains with solubles and 30% corn germ meal.

Stereotypic behaviors could be a major problem in individual gestation stalls due to animal welfare concerns and public perception (sustainability of production systems or acceptance of practices involving animals). Dietary fiber increases chewing, which limits intake by promoting the secretion of saliva and gastric juice, resulting in an expansion of the stomach and increased satiety. Sapkota et al. [42] recorded sows’ behaviors in nine different time points (starting at 0830 h). They included dietary fiber sources (resistant starch: 11%; sugar beet pulp: 27%; soybean hulls: 19%) in diets for sows and they found that the percentage of sows standing (71%) was highest in sows fed sugar beet pulp, while the percentage of resting was greatest for soybean hulls. However, bar chewing and nosing behaviors were not affected by fiber sources. The inclusion of 30% of wheat middlings and 15% of soybean hulls in sow’s diet results in an increased percentage of eating behavior (11.7 vs. 7.3), while percentage of sitting behavior was reduced in sows fed diets containing 30% distillers dried grains and 30% corn germ meal. However, lying, standing, drinking, sham-chewing, walking, and oral–nasal–facial behaviors were not affected by fiber sources [43]. While most stereotypic behaviors were not affected by dietary fiber sources, increasing resting behavior and reducing standing behavior was observed when sows were fed soybean hulls (19%) and resistant starch (5%), which may be indicative of decreased motivation of feeding and increased postprandial satiety. The soluble and insoluble ratios of dietary fiber sources (wheat middlings (30:1), soybean hulls (5 to 15:1), sugar beet pulp (5:1)) would be an important parameter to determine the physio-chemical characteristics of fiber [44,45,46]. The sugar beet pulp or soybean hulls are a moderately fermentable fiber, which contains both insoluble and soluble fiber components (high in pectin, cellulose, and hemicellulose) in a desirable ratio compared to other dietary fiber sources. Most studies above showed that supplementation with soybean hulls in the diet for sows reduced stereotypic behaviors (increased resting and decreased standing behaviors). This suggests that low solubility of fiber sources may be less effective in suppressing huger, resulting increased standing and eating behaviors in sows.

Discrepancies among the studies reported above may be due to dietary fiber source, fiber inclusion levels, or soluble and insoluble content (ratio) affecting outcome measures. However, the inclusion of dietary fiber in the diet for sows may reduce stereotypic behaviors because feeding fiber increases chewing, resulting in an expansion of the stomach and increased satiety due to accelerating the secretion of saliva and gastric juice. Based on the studies above, sugar beet pulp or soybean hulls could be potential candidates for dietary fiber sources to alleviate abnormal behaviors in sows.

The positive effects of sugar beet pulp or soybean hulls on sow behavior may relate to the character of fiber present. Both sugar beet pulp and soybean hulls are high in soluble fiber which is highly fermented by sows. Fermentation of dietary fiber increases the production of short chain fatty acids (SCFA) [47] which are readily absorbed through the intestinal wall into the blood. These SCFAs are available as an energy source during the interprandial period when glucose supply from the gut is diminishing. Because fermentation of dietary fiber requires more time relative to starch digestion, the SCFAs are available for a sustained period of time after each meal. De Leeuw et al. [48] demonstrated that a diet containing 45% sugar beet pulp improved the stability of plasma glucose and insulin in sows during the interprandial period from 3 to 12 h after a meal compared to sows fed a starch-based diet. Improved stability of plasma glucose and insulin levels was associated with a significant decline in postural changes of sows during the period from 5 to 12 h after a meal. Decreased postural changes presumably indicate that sows are more content and less restless. In a later study, De Leeuw et al. [49] demonstrated that fermentation of dietary fiber in the hindgut is more important than the increased gut fill associated with high-fiber diets in improving the stability of plasma glucose and insulin. Similarly, Brouns et al. [19] observed decreased peak postprandial concentrations of glucose and insulin and a prolonged elevation of serum insulin in pigs fed a diet high in sugar beet pulp. These data suggest that fermentable fibers are more effective in improving the welfare of gestating sows compared with non-fermentable fiber sources.

#### 2.2.3. Levels of Dietary Fiber

A logical approach to controlling stereotypic behaviors and excessive body weight gains during pregnancy is to provide sows with a high-fiber diet [14,50]. In agreement with the literature, by increasing levels of dietary fiber in the diets for sows, a corresponding reduction in nutrient digestibility and feed intake could be observed [51,52]. A higher intake of dietary fiber increases the luminal viscosity and water-binding capacity of digesta (soluble fiber), as well as passage rate in the small intestine (insoluble fiber). These physio-chemical properties of fiber may affect the feed intake and stereotypic behaviors of sows (Table 5).

The ingestion of high-fiber diets could decrease the feed intake because of the increased bulkiness of the diet. Moreover, in addition, increasing dietary fiber inclusion leads to a decrease in dietary energy intake, causing sows to increase feed intake to keep up with their energy requirements. However, results could be varied depending on dietary fiber sources and concentrations in the diet. Le Gall et al. [53] tested different levels of dietary fiber (low: 13%, medium: 21%, high: 31%, very high: 38%, total dietary fiber (TDF)) for sows. The diets were formulated with various dietary fiber sources including wheat, maize, barley, soybean meal, and a combination of fibrous ingredients (sugar beet pulp, wheat bran, maize bran, and soybean hulls). They found that increasing supplementation of dietary fiber linearly reduced feed intake. However, Sun et al. [54] found a linear increase in feed intake when sows were fed diets based on different levels of konjac flour (0, 0.6, 1.2, or 2.2%) during the lactation period. Loisel et al. [55] provided low-fiber (13%, TDF) and high-fiber (23%, TDF) diets to sows during the gestation period. In this study, however, a high-fiber diet fed to sows was expected to increase feed intake compared to a low-fiber diet, but no difference was observed in feed intake. Similarly, there was no difference in voluntary feed intake when sows were fed a high-fiber diet (15%, crude fiber) compared to the control diet during the lactation period (3%, crude fiber) [56]. Studies on these sows suggested that the fiber concentration below 15% of crude fiber may not affect feed intake. However, if the TDF concentration in the diet exceeds 23%, it is considered that the feed intake may be reduced for sows.

Ingestion of high-fiber diets has the potential to adversely affect energy and nutrient utilization due to the regulation of the digestive process and glycemic response [57]. Depending on the fiber source and consumed amount, different effects on nutrient digestibility could be promoted. Rijnen et al. [58] reported that group-housed sows were able to utilize energy from sugar beet pulp silage, which is high in soluble fiber, as efficiently as energy from starch. An important observation is that diets containing very high levels of fibrous feed ingredients can be just as digestible as high-starch diets for gestating sows. However, the study by Oelke et al. [59] showed that increasing the amount of total dietary fiber from soybean hull (0, 12, or 24%, TDF) in sow diets linearly decreased in the digestible energy and apparent total tract digestibility of dry matter, gross energy, crude protein, non-fibrous carbohydrates, and organic matter. Similarly, Calvert et al. [60] reported that using 50% and 95% alfalfa meal in the gestational diet decreased the digestibility of dry matter, protein, energy, and fiber components (neutral detergent fiber, acid detergent fiber, and cellulose) compared to 5% alfalfa treatment. In addition, the digestibility of dry matter, organic matter, nitrogen, carbohydrates, and energy decreased linearly with increasing fiber levels (from 13% to 38%, TDF), while the digestibility of crude fiber and acidic detergent fiber increased linearly [54]. This improvement is associated with the development of the digestive tract in sows, allowing increased degradation of dietary fiber fractions. Holt et al. [56] also reported that dry matter, energy, and nitrogen digestibility was decreased by feeding a high-fiber diet (15%, crude fiber) to sows. Based on the studies, nutrient digestibility has been reported to be negatively affected by the feeding of high-fiber diets to sows. The addition of fibrous feedstuffs (over 15%, TDF) in the diet for sows could reduce nutrient digestibility due to the increased digesta viscosity (soluble) or decreased digesta transit time (insoluble), which slow down the diffusion of the substrates and lead to less mixing time for digestive enzymes.

Few studies have compared the effects of different fiber sources and inclusion levels on the stereotypic behavior of sows. A diet containing 50% sugar beet pulp was more effective in reducing stereotypic behaviors and aggression in gilts than a diet containing 50% of mixed fiber sources (grass meal, wheat bran, oat hulls). Both high-fiber diets increased the time spent eating and resting while they reduced the time spent standing. These behavior patterns all suggest improved satiation of the sows [57]. Bergeron et al. [14] reported that sows fed very high-fiber diets (23%, crude fiber) spent less percentage of time in chain manipulation, vacuum chewing, nose rubbing, and object biting than sows fed a control diet (5%, crude fiber). Additionally, sows fed very high-fiber diets spent less time standing than sows fed the control diet. However, there were no differences in sitting and standing time between treatments. Holt et al. [56] suggested that feeding a high-fiber diet contained with soybean hulls (15%, crude fiber) increased the percentage of time for sitting and feeding activity, while it decreased the percentage of time lysing for sows during the gestation period compared to sows fed a control diet (3%, crude fiber). The increased feeding activity is associated with increased time spent standing and these behaviors seem to be driven by a physiological need for energy and gut fill (feed motivation). Guillemet et al. [61] tested a control diet (3%, crude fiber) and high-fiber diet (12%, crude fiber) which was formulated with wheat, barley, soybean meal, and mixed fiber. However, no differences were observed in nesting, resting, sitting, standing, ventral, and lateral recumbency behaviors in sows during parturition. Bernardino et al. [62] evaluated sows’ behaviors in two feeding periods (one hour before and one hour after feeding) with low-fiber (3%, crude fiber) and high-fiber diets (13%, crude fiber) during the gestation period. They did not find any differences in feeding motivation and any behaviors for duration or frequency (sleep, lysing, standing, sham chewing, rooting floor, licking floor, fence and gate interaction, interacting with mats, and vocalization) between the treatments. However, sows fed the low-fiber diet showed longer and frequent sham-chewing and licking floor behavior before feeding compared to after feeding treatment. DeDecker et al. [63] used two different floor spaces (1.7 m^2^ and 2.3 m^3^) and diets (control: 3 and 9%, acidic detergent fiber (ADF) and neutral detergent fiber (NDF) and fiber: 17 and 28%, ADF and NDF) for sows during gestation periods. They concluded that the sows that were fed the fiber diet and raised at 1.7m^2^ performed lower oral–nasal–facial and sham-chewing behaviors compared to the sows that were fed the control diet and raised at 1.7m^2^. Consequently, definitive recommendations on the best fiber source or optimal levels of fiber to eliminate behavioral vices are difficult. However, levels of dietary fiber in diets between 12 and 15% (crude fiber) for sows may not affect the stereotypies.

**Table 5 animals-13-00879-t005:** Results of studies evaluating the effect of various levels of dietary fiber on feed intake, nutrient digestibility, and behaviors of sows.

Feeding Period	Fiber Sources or Treatments	Ingredient or Fiber Concentrations	Main Results	References
Gestation	Low fiber vs. high fiber ^a^	13 and 23%, TDF levels	No effect on feed intake, BW, and backfat thickness	Loisel et al. [55]
Gestation	Alfalfa meal	5, 50, or 95%	Decreased dry matter, fiber components, protein and energy digestibility with increasing alfalfa meal	Calvert et al. [60]
Gestation	Control vs. SBP vs. MFS ^b^	5, 13%, or 14%	SBP and MFS increased time spent eating and resting and decreased spent standing	Danielsen and Vestergaard [57]
Gilts	Sugar beet pulp silage	0, 10, 20, or 30%	Increased energy digestibility	Rijnen et al. [58]
Non gestation and lactation	Low vs. medium vs. high vs. very high fiber ^c^	13, 21, 31, or 38%, TDF levels	Linear decrease nutrients (DM, OM, N, CHO, and energy) digestibility and daily food intake, but linear increase in CF and ADF digestibility	Le Gall et al. [53]
79 to 113 days of gestation	Soybean hull	0, 12, or 24%	Linear decrease nutrients digestibility	Oelke et al. [59]
			No effect on feed intake and BW	
Gestation	Control vs. high fiber vs.very high fiber ^d^	5, 18, or 23%, CF	Sows fed very high-fiber decreased time performing stereotypies	Bergeron et al. [14]
Post weaing to lactation	Control vs. soybean hulls	3% or 15%, CF	Increased percentage of time sitting and feeding activity and decreased percentage of time lying during gestation period and nutrient digestibility, no effect on feed intake during lactation period	Holt et al. [56]
Gestation and lactation	Control vs. high fiber ^e^	3% or 12%, CF	No effect on activities and postures in sows	Guillemet et al. [61]
Gestation	Control vs. high fiber ^f^	3 and 9% vs. 17 and 28%, ADF and NDF ^g^	Decreased oral–nasal–facial and sham-chewing behaviors	DeDecker et al. [63]
Gestation and lactation	Konjac flour	0, 0.6, 1.2, 2.2%	Linear decrease non-feeding oral behavior, linear increase in feed intake during lactation period	Sun et al. [54]
Gestation and lactation	Low fiber vs. high fiber ^h^	3% or 13%, CF	No effect on feed motivation test and any behaviors for duration or frequency	Bernardino et al. [62]

^a^ Low fiber was based on barley and wheat, high fiber was based on barley, wheat, soybean hulls, wheat bran, sunflower meal, and sugar beet pulp. ^b^ SBP: sugar beet pulp, MFS: mixture of green grass meal, wheat bran, and oat hulls. ^c^ Dets were based on wheat, maize, barley, soybean meal and a combination of fibrous ingredients (sugar beet pulp, wheat bran, maize bran, and soybean hulls). ^d^ High-fiber and very high-fiber diets were formulated with oat hulls and alfalfa meal. ^e^ Control diet was based on wheat, barley, and soybean meal, high fiber was based on wheat, barely, and fiber rich feedstuffs (sugar beet pulp, soybean hulls, wheat bran). ^f^ Control diet was based on soybean meal and high-fiber diets was based on wheat middlings and soybean hulls. ^g^ ADF: acidic detergent fiber, NDF: neutral detergent fiber. ^h^ Soybean hulls was the main fiber source.

In most studies that demonstrated a significant effect of dietary fiber on stereotypic behaviors, fibrous ingredients were included at high levels (over 20%, TDF). These high inclusion rates for fibrous ingredients significantly decrease the bulk density of feed which creates nearly insurmountable problems with conveyance of mash feed through augers and storage bins and difficulty in thoroughly mixing diets. The high inclusion rates for fibrous ingredients make it difficult to produce high quality pellets for pelletized feed.

These handling problems make many diets proven to be effective in modifying sow behavior impractical for commercial production. Holt et al. [56] addressed this practical problem by formulating a high-fiber, mash diet based on corn, soybean meal, and 40% soybean hulls, which readily flowed through commercial feed delivery systems. Sows receiving the high-fiber diet were allotted 20% more feed (2.42 vs. 2.03 kg/day) than control sows each day in an attempt to compensate for the energy-diluting effects of soybean hulls. The high-fiber diet had no effect on reducing the occurrence of stereotypic feeding behaviors in sows for 3 h around feeding time. They used sows that were in their second or greater pregnancy and experienced at least one gestation period housed individually in stalls. These sows may have been unresponsive to the effects of dietary fiber because their feeding behaviors were already sensitized by previous housing in stalls. These older sows may have developed a consistent repertoire of behaviors, including stereotypic behaviors, that were elicited by feeding. Consequently, regardless of diet composition, sows expressed these behaviors as a result of feeding. Lawrence and Terlouw [12] referred to this phenomenon a “channeling” of behaviors. Similarly, van der Peet-Schwering et al. [64] noted that stereotypic behaviors increase as sows age and that high-fiber diets are less effective in reducing stereotypies in older sows.

Differences between the reported works and previous studies may be related with quality of fiber, inclusion of fiber level (adding oil is necessary if we used a greater amount of fiber that affect palatability and feed intake), multiparous sows (age and experience of farrowing), format of diet (mash or pellet), and floor space for sows. The abovementioned studies reported on diverse types of dietary fiber that mentioned “high fiber or very high fiber”, which contained varying inclusion levels of fiber sources. The scientific literature is clear that specific formulations of high-fiber diets can decrease the stereotypic behavior of gestating sows. Based on the studies, the inclusion of high levels of sugar beet pulp (13–30%) or soybean hulls (13%) seems to provide a consistent reduction in stereotypic behaviors and increases the satiety of sows [57,58,59,62,63,64].

## 3. Other Approaches to Decrease Stereotypies

Feeding management practices may influence the expression of stereotypic behaviors. We theorized that offering sows two meals per day instead of one might provide more feeding opportunities for sows throughout the day which would decrease their need to exhibit stereotypic behaviors. However, Holt et al. [56] found that total time sows expressed stereotypic behaviors each day increased when the daily feed was divided in two meals compared to a single daily meal. The channeling of feeding-associated behaviors may explain why twice daily feeding was not beneficial. Manu et al. [65] offered one meal, two meals, and three meals per day for sows to evaluate how feeding frequency affects cortisol response and behaviors (food anticipatory activity, total feed activity, and indication of hunger). They suggested that no difference was found between the sows fed two or three meals because all sows had a similar energy intake. However, the counts of food anticipatory activity, total feeding activity, and indication of hunger were greater for sows fed three meals compared to sows fed one or two meals. These results indicated that feeding frequency did not provide sufficient gut fill, which may not reduce the stereotypic behaviors in sows. In support of this theory, Poulopoulou et al. [66] reported that feeding sows three meals per day increased feed intake during lactation period, while a reduced degree of shoulder lesions was observed in sows fed three meals per day compared to sows fed two meals per day. Therefore, frequent feeding, regardless of the fiber content of the diet, is not likely to decrease the occurrence of stereotypic behaviors in sows that already display these abnormal behaviors. However, more frequent feeding may effectively reduce the development of stereotypic behaviors in young sows.

Feeding motivation is the product of internal physiological cues and external stimuli [12]. Consequently, one may be able to enhance satiation of the sow and decrease the occurrence of stereotypic behaviors with a combination of high-fiber diets and some specific metabolic modifier. Brouns et al. [19] reported diets high in fermentable fiber from sugar beet pulp blunted the postprandial response in insulin and glucose and increased circulating levels of short-chain fatty acids. Presumably, these metabolic changes were present in sows that exhibited an enhanced level of satiety after consuming high-fiber meals. Possibly, some intermediary metabolite normally responsible for the control of feed intake could be manipulated by diet composition to mimic conditions of a large meal. If so, sows could receive limited amounts of a diet moderately high in fiber and achieve the same level of satiation as sows fed large quantities of diets very high in fiber. This approach, if successful, would minimize the practical problems of very high-fiber diets while achieving the reduction in stereotypic behaviors realized when feeding diets very high in fiber.

## 4. Conclusions

There are proven methods of controlling the hunger and associated stereotypic behaviors in pregnant sows. However, these methods have important limitations, which prevent them from being rapidly adapted to commercial production systems. More detailed research into the mechanisms responsible for the decline in stereotypic behaviors caused by high-fiber diets will provide solutions to these practical challenges. In the interim, sows should receive diets that contain over 13% crude fiber and contain ingredients with high concentrations of fermentable fiber if a goal is to minimize expression of stereotypic behaviors. The beneficial effects of high-fiber diets will only be realized if the sow’s nutrient requirements for maintenance, growth, and reproduction are met. Feeding diets in meal rather than pelleted form may be more effective in reducing stereotypies realizing that feed handling problems are magnified.

## Figures and Tables

**Table 1 animals-13-00879-t001:** Definition of behaviors for behavioral observation of pregnant sows ^a^.

Behaviors	Definition
Sham chewing	Continuous chewing without the presence of visible food in the oral cavity
Rooting the floor or feeder	Snout touches the ground followed by head
Standing	Body supported by the four limbs
Lying ventrally	Lying with the belly on the ground with all the limbs under the body
Lying laterally	Lying sideways, with all the limbs extended laterally
Licking the floor or feeder	The tongue touches the floor and is followed by movements with the head
Interacting fence or gate	Biting or nibbling the fence wire or gate
Interacting with mats	Snout or tongue touches mats followed by head movements
Bites	Bite on any parts of the body (tail, vulva, ear, body)
Facing	Face to face, with a fixed view to the other animal
Pushing	Pushing another animal using the head or the muzzle
Vocalization	Sound emission emitted by the animal
Belly nosing	Snout movements to stimulate the milk flow

^a^ Reprinted/adapted with permission from Ref. [9]. 2004, Zonderland et al.; Reprinted/adapted with permission from Ref. [10]. 2020, Tatemoto et al.

## Data Availability

Not applicable.

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
