# Peer review of "The Role of Dietary Fiber in Improving Pig Welfare"

_animals, 2023, doi:10.3390/ani13050879_

Round 1

Reviewer 1 Report

The review manuscript of Do et al. critically reviews the scientific information available on the application of fiber in the diet of the pregnant sow for improving animal welfare. The language is clear and easy to understand. The hypothesis is well explained. The manuscript has some interesting recommendations. The structure of the manuscript is appropriate with some minor changes that may be undertaken to improve its presentation.

I have following observations to improve its quality as-

L13: sow being hunger or being hungry, plz check

Keywords: may add gestation

L189: public perception, please clear it

As sugar beet and soybean hull topic having good information, so may be grouped under a new subheading

Table 3: may be presented in more suitable way may be in figure form

Author Response

Response to Reviewer 1 Comments

Point 1: L13: sow being hunger or being hungry, plz check.

Response 1: I changed it to hungry.

Point 2: Keywords: may add gestation

Response 2: I added gestation at Keywords.

Point 3: L189: public perception, please clear it

Response 3: I added the meaning of public perception.

Point 4: As sugar beet and soybean hull topic having good information, so may be grouped under a new subheading

Response 4: Thank you for your comments. Yes. We found positive results when sows fed diet with sugar beet pulp or soybean hull from a few papers, but degree of solubility or fementability of these fiber sources could be varied depending on the enviornment, processing methods, e.t.c. So, if we grouped these under a new subheading, readers should focus on that subheading. In my opinion, I would keep this format.

Point 5: Table 3: may be presented in more suitable way may be in figure form

Response 5: Thank you for your recommendation, but I would keep this table form because table form is more easier to understand the characteristics of dietary fiber.

Reviewer 2 Report

Supplement in the text and summary (conclusion) with the procedure in organic farming, where grass or corn silage is used, and green fodder in the summer.

It is also worth quoting an interesting publication by Selene Jarrett and Cheryl J. Ashworth (2018) - Journal of Animal Science and Biotechnology.

Detailed remarks:

Line 18 - add dried grass

Line 24 - better – diets

Line 100-106 - Please complete the information on the detergent fiber: NDF, ADF and ADL and provide the composition of these fractions.

Line 160 or on another line - Please complete information on the impact of other fibrous feeds, e.g. dried grass.

Author Response

Response to Reviewer 2 Comments

Point 1: Line 18 - add dried grass.

Response 1: Thank you for your comment. I added it.

Point 2: Line 24 - better – diets

Response 2: Thank you for your comment. I changed it.

Point 3: Line 100-106 - Please complete the information on the detergent fiber: NDF, ADF and ADL and provide the composition of these fractions.

Response 3: Thank you for the suggestion. Yes. I added more information on the detergent fiber content.

Point 4: Line 160 or on another line - Please complete information on the impact of other fibrous feeds, e.g. dried grass

Response 4: Thank you for your kind suggestion. I tried to find more information about forage grasses for sow, but there were only few studies. Also, using forage grasses would be associated with organic pig farming (group housing) or for ruminant animals, but this review paper focused on sows being individual housing (stall) condition. and I reviewed some studies about sugar beet pulp silage or wheat straw for sow’s diet. In my opinion, If I add more information about forage grasses, then concept of this review paper should be changed.
